# Regional Suicide Rate Change Patterns in Korea

**DOI:** 10.3390/ijerph17196973

**Published:** 2020-09-24

**Authors:** Byung-sun You, Kyu-hyoung Jeong, Heeran J. Cho

**Affiliations:** 1Policy Research Team, Gyeonggi Welfare Foundation, Seoul 16207, Korea; sunmagnolia@hanmail.net; 2Policy Development Team, Nowon-gu Office, Seoul 01689, Korea; 3Department of Health Administration, Yonsei University, Seoul 26493, Korea

**Keywords:** typology study, suicide in Korea, growth mixture model, latent growth model, multinomial logistic analysis

## Abstract

*Background:* Korea had the highest suicide rate among OECD countries for the 10 years leading up to 2016; however, the suicide rate in Korea has declined since 2010, after policy-driven interventions were implemented. *Methods:* Suicide rates from all of the 229 cities, counties, and districts in Korea were reliably estimated from the period 2010 to 2017, and data were examined by Stata 14.0 and M-plus to identify regional suicide rate change patterns by latent growth modeling. The dependent variable is the suicide rate, and independent variables as characteristics of the various districts are the region (cities, counties, and autonomous districts), proportion of elderly residents, financial independence rate, establishment of mental health and welfare centers, and number of social welfare facilities. Results: Three suicide rate change patterns were identified: ‘average’, ‘precipitous drop’, and ‘high level’. Two of the three patterns exhibit features that are markedly different to the national data as a whole, and the three patterns appear across the 229 cities, counties, and districts of Korea. Some of the determinant factors have been postulated here. While a high proportion of elderly residents in a given area is a significant indicator that the suicide rate will increase, having a large elderly population in combination with an increased number of social welfare facilities centers appeared to show a discrete pattern of suicide rate reduction when compared with average national data. *Conclusions:* Policy-driven interventions should be planned and implemented by central and local governments in conjunction, by considering regional characteristics to decrease local suicide rates more effectively.

## 1. Introduction

Suicide is a major public health concern worldwide, representing approximately 800,000 deaths every year [1]; it accounted for 1.4% of total global mortality and was the 18th leading cause of death in 2016 [2]. As suicide is a preventable cause of death, different countries are making efforts to reduce its effect by implementing national suicide strategies [3]. Higher social expenditure in welfare provision has been associated with suicide-preventive effects in Europe in some instances [4], and government-led suicide prevention programs in a number of OECD countries were effective among youths and the elderly [5].

Factors affecting suicide rate include both promoting and protective factors at different levels. These levels include: individual, family, community, and society [6,7,8,9]. Many factors are associated with suicide and affect suicide rate, ranging from the unemployment rate and financial stress [10,11] to mental health and social welfare support [4,12]. Mental support was dramatically effective for Finnish elderly in preventing suicide [13], and social welfare services played an important role in preventing suicide mortality in European countries during times of economic recession [4]. Suicide mortality varies by age, sex, urbanicity and regional factors [12,14,15]; the suicide rate among people over 65, males, and people residing in rural areas has remained higher in many countries. Suicide rate varies among countries, and it also varies between regions within a country (e.g., [14,16,17,18,19]). While the suicide rate is influenced by socio-economic factors [20,21,22], it varies regionally due to circumstances including residential and regional economic status [12]. 

Korea (population: 51,780,579 in 2020) had the highest suicide rate in the OECD for the majority of the last 13 years [23]. The suicide rate in Korea increases with age, making a pronounced rise after 70, and becoming the most severe among people over 80 [24]; while suicide rate changes are barely observed among teens and people in their 20s and 30s [25]. The overall suicide rate in Korea began to increase after 1992 (8.3 per year per 100,000 people). It took a sharp turn upwards in 1998 (18.6 per year per 100,000 people) after the International Monetary Fund (IMF) Crisis and again in 2009 (31.0 per year per 100,000 people) after the Global Financial Crisis. The suicide rate has decreased sharply from 2010 (31.2 per year per 100,000 people) to 2017 (24.3 per year per 100,000 people) [23,24]. This study explored the data since 2010, when the suicide rate in Korea began to decline and explored regional changes in categorization via the strength of the statistical method Growth Mixture Model (GMM). The purpose of this study is to derive the regional suicide rate change patterns across all of the 229 cities, counties, and districts in Korea by exploring regional characteristics, including the proportion of elderly residents, financial independence rate, the social welfare infrastructure, and the establishment of mental health and welfare centers. Furthermore, it is necessary to determine the characteristics of each suicide rate change pattern and to identify the crucial factors for the regional patterns.

## 2. Method

### 2.1. Data

The suicide rate data in Korea were utilized from Statistics Korea. The raw data of Statistics on the causes of death between 2010–2017 were obtained by remote access to the Micro Data Integrated Service (MDIS) of Statistics Korea (Statistics Korea, Daejeon, Korea). The number of suicides in cities, counties, and autonomous districts was calculated based on intentional/self-inflicted death as denoted by codes (X60-X84) in the 5th Korean standard classification of disease and causes of death, which was modified from the WHO’s International Classification of Disease codes (10th revision). Each city, county, and autonomous district was coded into individual cases. In this study, all of the 229 cities, counties, and districts (76 cities, 84 counties, and 69 autonomous districts) where suicide rates could be reliably and accurately estimated from 2010 to 2017 were studied.

Specifically, they were classified from one special city, six metropolitan cities, and six provinces. Cities whose population is over 500,000 are termed autonomous districts. Specifically, there are 25 districts of Seoul special city, 1 county and 15 districts of Busan metropolitan city, 1 county and 7 districts of Daegu metropolitan city, 2 counties and 8 districts of Incheon metropolitan city, 5 districts of Gwangju metropolitan city, 5 districts of Daejeon metropolitan city, 1 county and 4 districts of Ulsan metropolitan city, 27 cities, 4 counties, and 20 districts of Gyeongi province, 7 cities and 11 counties of Gangwon province, 3 cities, 9 counties, and 2 districts of Chungcheongbuk province, 8 cities, 7 counties, and 2 districts of Chungcheongnam province, 6 cities, 9 counties, and 2 districts of Jeollabuk province, 5 cities, and 17 counties of Jeollanam province, 10 cities, 13 counties, and 2 districts of Gyeongsangbuk province, 8 cities, 10 counties, 5 districts of Gyeongsangnam province, and 2 cities of Jeju province.

### 2.2. Variables

#### 2.2.1. Dependent Variable: Suicide Rate

From 2010 to 2017, the suicide rate was calculated for every 100,000 people in each city, county, and district through the formula ‘(number of suicides/mid-year population) × 100,000′ using cause of death statistics and resident registration-related population data. 

#### 2.2.2. Independent Variables: Characteristics of Districts

Cities, counties, and districts were measured on five dimensions: region (cities, counties and autonomous districts), proportion of elderly residents, financial independence rate, establishment of mental health and welfare centers (centers established by each city, county, and district to prevent suicide and to provide suicide risk intervention), and number of social welfare facilities (facilities established by each city, county, and district to provide social welfare, to people who have or may have social problems). These variables were found to affect suicide rate based on the previous studies [4,12,13,16,17,18,19].

For the region variable, the value was set to zero for city districts and one for county districts as a dummy variable. The proportion of elderly, aged 65 or older, to the total population was set through census data from Statistics Korea. Financial independence rate was measured as the proportion of initiatives implemented and administered by each city, county or district’s autonomous office (e.g., municipal office, county office, district office) and funded internally rather than those that were funded through external sources, obtained through the data of the local financial integration disclosure system. It was calculated using the formula below.
(1)$$Financial\;Independent\;Rate\left(\%\right)=\left\{\frac{Local\;tax\;revenue+non\text{-}tax\;revenus-municipal\;bond}{General\;accounts\;tax\;revenue}\right\}\times 100$$

Mental health and welfare centers were established to manage patients with chronic mental diseases and to promote the mental health of the community, while social welfare facilities were established to provide social welfare services to people who have social problems or potential social problems, and specifically for elderly, low-income, disabled, infant, juvenile, and adolescent people.

Whether a mental health and welfare center was set up was set to 0 if it was not established and 1 if it was established, using the Ministry of Health and Welfare’s 2018 Mental Health and Welfare Report. While there are a number of cities, counties, or districts where health and welfare centers are established, there are still areas where the centers have not been established at all, so this variable is coded with 0 or 1. The number of social welfare facilities was set as the number of social welfare facilities per 1000 people using the data of the social welfare facilities information system. 

### 2.3. Statistical Analysis

Analysis methods and procedures for resolving research problems in this study are as follows. Stata 14.0 and M-plus 8.0 programs were used for data handling and model analysis. First, a descriptive statistical analysis was conducted to identify the subject of the study and the characteristics of the main variables. Second, to estimate the trajectory of the overall suicide rate, a latent growth model (LGM) was implemented, assuming that all the data were from a single, homogenous group. The latent growth model is a method of analyzing the average change pattern of individuals with a single population through repeated measured observations. Furthermore, in order to determine the goodness of the model, and Tucker and Lewis Index (TLI), Comparative Fit Index (CFI), and Root Mean Square Error of Approximation (RMSEA), considering the model’s degree of simplicity and the fact that it was not sensitive to the size of the sample. 

Third, GMM was implemented to distinguish the suicide rate change patterns. It was also implemented to overcome the limitation that the subject of the data was previously assumed to be one group, substituting it with an assumption that the data came from several sub-populations to determine the number of latent groups with different trajectories. Therefore, in the growth mixture model, the variance representing individual differences in the trajectory, including the overall shape of the development trajectory, and the effect of the factors explaining it, are estimated to be different latent classes. In other words, the purpose of the growth mixture model is to estimate the number of unobserved latent groups from repeated measured longitudinal data to reveal different trajectories for each group.

The *p*-values of Akaike Information Criterion (AIC) [26], Baysian Information Criterion (BIC) [27], Sample-size Adjusted BIC (SABIC) [28], Entropy and Bootstrap Likelihood Ratio Test (BLRT) [29] were used to determine the number of change patterns in the GMM, identifying each sub-groups’ characteristics. Fourth, multinomial logistic analysis was conducted to identify the crucial factors determining the pattern of suicide rate change by each city, county, and district.

## 3. Results 

### 3.1. Descriptive Statistics

Descriptive statistical analysis was conducted to identify the characteristics of the main variables (Table 1). Suicide rates (reported in suicides per year per 100,000 people) have continuously decreased from 2010 (M = 37.70 and SD = 12.84) to 2017 (M = 27.26 and SD = 7.57). A total of 147 city districts (64.2%) and 82 county districts (35.8%) were identified. For mental health and welfare centers, 150 were established before 2010 (65.5%) and 79 were established after 2011. The average proportion of elderly residents was 15.56% (SD = 7.39), residents (SD = 0.10).

### 3.2. Analysis of Latent Growth Modeling 

LGM was implemented to identify the trajectory of the overall suicide rate before proceeding with the GMM (Table 2). For this, each city, county, and district is assumed to be an equal group, and to identify the optimal change pattern, the goodness of model fit is tested by analyzing a no-change model, linear change model, and quadratic function change model, respectively, in comparison.

The goodness of fit for the no-change model regarding the suicide rate was χ^2^ = 491.523 (*p* < 0.001), CFI = 0.471, TLI = 0.564, RMSEA = 0.242, representing that it does not satisfy the goodness of fit. The goodness of fit for the linear change model was χ^2^ = 61.923 (*p* < 0.001), CFI = 0.966, TLI = 0.961, RMSEA = 0.073, which was shown to be better than the goodness of fit for the quadratic function change model (χ^2^ = 61.923 (*p* < 0.001), CFI = 0.966, TLI = 0.961, RMSEA = 0.073), CFI = 0.950, TLI = 0.988, RMSEA = 0.75); the linear change model was selected accordingly.

### 3.3. Determining the Number of Patterns According to the Goodness of Model Fit

As the linear change model was shown to illustrate the trajectory of suicide rate well through the LGM, the linear change model was also accordingly estimated in the GMM (Table 3). In addition, the *p*-values of AIC, BIC, SABIC, Entropy and BLRT, as well as more than 5% of all cases, were used to determine the values for suicide rate change patterns for each city, county, and district.

Each autonomous district (city, county, district)’s suicide rates were generally shown to be decreased as estimated pattern numbers of AIC, BIC and SABIC were increased. Entropy was higher for the group 3, at 0.092 with three classes (patterns), than other groups with different numbers of classes. BLRT was shown to be statistically significant at a level of *p* < 0.001 for groups 2 and 3, with two and three classes, respectively. 

Additionally, in order to see if every pattern (class) is bigger than 5% of the overall sample size, only groups two and three with two classes and three classes were acceptable. Considering the goodness of the model fit synthetically, it is the most suitable for the suicide rate to be shown with three classes, and group 3 with three classes was selected as the final model. The three pattern’s sample sizes were 168 cases (73.4%), 45 cases (19.6%) and 16 cases (7.0%), respectively.

The suicide rate by district was classified into three classes; each class was named to reflect the initial level of suicide rate and the characteristics of the aspect of change (Table 4 and Figure 1). The first class was named ‘average’ pattern, as the pattern appeared to be similar to the average national level of change in suicide rate. The second class was named ‘precipitous drop’ pattern, as initially high levels of suicide were shown to drop sharply over time. The third class was named ‘high-level’ pattern, as it showed that a high level of suicide rate was sustained. 

### 3.4. Crucial Factors of Suicide Rate Change Patterns 

A multinomial logit analysis was conducted to verify the crucial factors affecting differences in the suicide rate change classes. Based on the characteristics of each city, county, and district, crucial factors for determining the ‘precipitous drop’ pattern as compared with the ‘average’ as a standard are the proportion of elderly residents (Coef. = 0.114, *p* < 0.05), and the number of social welfare facilities (Coef. = 7.021, *p* < 0.01). This means that as the proportion of the elderly residents is higher, and the number of social welfare facilities is higher, taking into account that these factors do not affect each other, there is a higher chance that the city, county, or district would show a ‘precipitous drop’ pattern than ‘average’. On the other hand, region, financial independence rate, as well as the establishment of mental health centers and social welfare facilities, did not significantly affect the suicide rate change pattern.

The major factor that distinguished a ‘high level’ pattern from an ‘average’ pattern was the number of social welfare facilities (Coef. = 6.674, *p* < 0.05). This means that if the number of social welfare facilities is higher, it is more likely for the city, county, or district to have a ‘high level’ than ‘average’ pattern. On the other hand, it was found that region, proportion of elderly residents, financial independence rate, as well as establishment of mental health and welfare centers did not significantly affect the suicide rate change pattern.

Estimating the Marginal Effect of the crucial factors for the suicide rate change pattern, it was found that an increase in the proportion of elderly residents by 1% increased the probability of exhibiting the ‘precipitous drop’ pattern over the ‘average’ pattern by 1%. For the number of social welfare facilities, as the number increased per 1000 people, the probability of exhibiting the ‘precipitous drop’ pattern rather than the ‘average’ pattern increased by 54.1% and the probability of exhibiting the ‘high level’ pattern rather than the ‘average’ pattern increased by 20.1%.

## 4. Discussion 

In this study, three suicide rate change patterns in the districts across Korea are identified. First, according to our methodology, there are three suicide rate change patterns in Korea in recent years: ‘average’, ‘precipitous drop’, and ‘high level’. A majority of districts (73.4%) display the ‘average’ pattern, 19.6% display a ‘precipitous drop’ pattern, and 7.0% exhibit a ‘high level’ pattern. While Korea has one of the top suicide rates in the OECD, the suicide rate in Korea decreased from 37.70 per year per 100,000 people in 2010 to 27.26 per year per 100,000 people in 2016, and this was viewed positively by WHO. However, it is revealed that 19.6% of districts experienced a reduction in suicide rates that far exceeded the national average. Likewise, a more complex picture is revealed when we examine Korea on a district-by-district basis, as shown above. Since different rate change patterns have been observed across the districts (cities, counties, districts) of Korea, it may be more appropriate to implement policy-driven suicide prevention programs at the district level. 

Second, the proportion of elderly residents and number of social welfare facilities are significantly linked with the ‘precipitous drop’ pattern. This finding indicates that the districts that have a high proportion of elderly residents and a high number of social welfare facilities also display a sharp decrease in the suicide rate. In other words, it suggests that when there are more public supports in a district with a higher proportion of elderly residents (an at-risk population), the suicide rate may drastically improve. 

Third, the districts with higher numbers of social welfare facilities were linked with a pattern of suicide rate change that remained at a ‘high level’, except in districts where the proportion of elderly residents was relatively high. This means that districts with a high number of social welfare facilities tended to exhibit the ‘high level’ pattern. The meaning of having many social welfare facilities in a city, county, or district is that there are many vulnerable groups in the local vicinity, which causes the suicide rate to increase, but this is only an inference. More detailed analysis is needed for a more detailed discussion.

A high number of social welfare facilities is related to the ‘high level’ as well as the ‘precipitous drop’ patterns. The higher the number of social facilities, the higher the probability of a district exhibiting either a ‘precipitous drop’ or ‘high level’ pattern. This means the number of social welfare facilities may decrease the suicide rate or merely maintain the rate, depending on the needs and vulnerability of the district population. In other words, the number of social welfare facilities may not be the factor that drops the suicide rate, and this implies that to decrease the suicide rate, simply increasing the number of social welfare facilities may not be the best solution.

Forth, the establishment of mental health and welfare centers did not show a strong enough relationship to classify the district pattern. It was discussed that suicide rates of all levels were inversely related to the number of mental health facilities in Greece [30], and that spending on mental health does not significantly affect the suicide rate [31]. The establishment of mental health and welfare centers also did not have an effect on the regional suicide rate change pattern [31]. Financial strain can influence the suicide rate indirectly [32], but it was also not a significant determinant factor affecting the suicide rate change patterns for districts.

Lastly, district type was also not an effective determinant factor. Kowalski et al. and Middleton et al. discovered that the gap in suicide rates in urban and rural areas is not noticeable [33,34]. It was observed that the suicide rate in urban areas has increased and exceeded the suicide rate in rural areas as a result of urbanization [33,34]. On the other hand, it was observed that suicide rates in rural areas are higher than urban areas as a result of lower socio-economic status, insufficient health and welfare infrastructure, and relatively easy access to pesticides (as a suicide tool) in rural areas [35,36,37] 

To implement suicide prevention policies, district-specific initiatives may be more effective as each district has distinct characteristics, and local governments can better predict how their constituents will respond to public initiatives. The central and local government should first determine the specific population make-up and needs of the target district. As the number of social welfare facilities was found to be higher in the districts with a high suicide rate in groups apart from the elderly, rather than increasing the number of facilities, understanding the district’s characteristics may be crucial. Accordingly, strategies are required to promote projects related to suicide prevention differently for the various district types.

Attempting to investigate characteristics of local suicide rates based on the proportion of elderly residents, financial independence rate, establishment of mental health and welfare centers, and the number of social welfare facilities has revealed the importance of establishing a suicide prevention plan taking regional characteristics into account. This study is intended to serve as a springboard for the central government to set the direction related to suicide prevention policies for each district and prioritize them accordingly. In addition, each district government will be able to clearly develop more effective ways of decreasing the suicide rate for the region, and this will help them set the direction when carrying out policies and strategies such as the suicide prevention plan.

Furthermore, since suicides tend to have a geographic pattern of “hotspots” and a pattern of “lethal methods” [38], centers like mental health and welfare centers to applying local characteristics for suicide prevention could decrease suicide rates more dramatically and effectively. Considering the place and method of suicides to prevent suicide and to provide suicide risk intervention at the right time and place would bring about more effective results. 

## 5. Conclusions

Suicide is a preventable cause of death with many nebulous and interacting determinant factors. No single intervention, or single set of interventions, will be equally effective wherever they are applied. According to our findings, the effectiveness of interventions is highly dependent on the characteristics of the district and local population in which they are being implemented. Specifically, at-risk populations, such as the elderly, respond to certain types of support differently than the general population, and their suicide rate change pattern alters accordingly. Our findings suggest that policy-driven suicide prevention interventions should be applied on a district level, accounting for the unique characteristics of each district. 

### Limitations

While this study carries significant implications for its accomplishment of creating a classification system for suicide rate change patterns on the local district level, this study did not divide the populations under investigation by age group and gender. For further studies, narrowing down the groups within each district would also be beneficial. 

Moreover, a more precise approach is necessary to reveal the factors that have more significance on the district level; specifically, the gap between the urban and rural suicide rate. Likewise, regional factors are mixed and not clearly revealed. Clarifying the mixed factors on the district level would improve further studies. 

Furthermore, as this study was mainly based on secondary quantitative research data, control of variables is not easy, and the independent variables of this study are a mixture of general factors and specific facilities. This point can be difficult to account for when interpreting the results of this study, so, in subsequent studies, if the facilities are defined and categorized in greater detail on a micro level, a clearer analysis result can be derived.

Lastly, the findings could be further developed and elaborated on by conducting qualitative research. For instance, the relationship between the number of social welfare facilities and changes in suicide rates could be understood and interpreted from a qualitative approach. This would supplement the current understanding of the hidden meaning behind the pattern change in the suicide rate in Korea and allow to interpret it more accurately.

## Figures and Tables

**Figure 1 ijerph-17-06973-f001:**
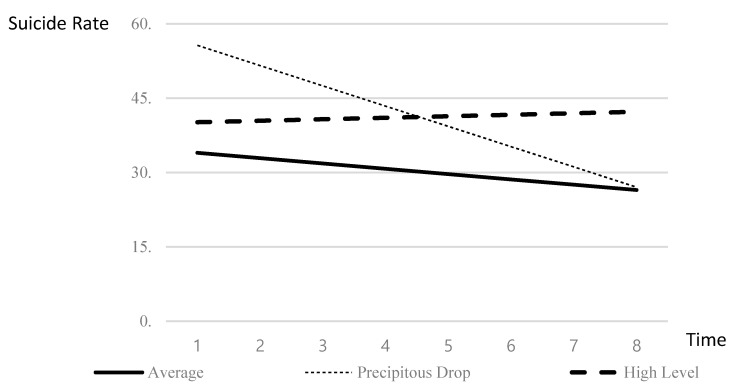
Estimates of the Initial Value and the Rate of Change by Suicide Rate Change Pattern.

**Table 1 ijerph-17-06973-t001:** Descriptive Statistics of Major Variables.

Classification		M	SD
Suicide Rate	2010	37.70	12.84
	2011	37.93	12.44
	2012	33.32	10.96
	2013	32.78	9.97
	2014	30.61	9.53
	2015	30.22	8.44
	2016	29.07	8.52
	2017	27.26	7.57
Characteristics of Each District	Proportion of elderly residents	15.56	7.39
	Financial Independence Rate	28.65	14.88
	Number of Social Welfare Facilities	0.15	0.10

The value is per 100,000 per year.

**Table 2 ijerph-17-06973-t002:** Fitness for Latent Growth Modeling of Suicide Rates.

Model	Χ^2^	*df*	CFI	TLI	RMSEA
No Change Model	491.523 ***	34	0.471	0.564	0.242
Linear Change Model	68.750 ***	31	0.966	0.961	0.073
Quadratic Function Change Model	61.923 ***	27	0.950	0.948	0.075

*** *p* < 0.001.

**Table 3 ijerph-17-06973-t003:** Growth Mixture Model Fitness.

Group	Goodness of Model Fit Index					Sub-Group Classification
	AIC	BIC	SABIC	Entropy	BLRT (*p*-Value)	Class Size (%)
1	12,737.408	12,782.046	12,740.845	-	-	-
2	12,695.064	12,750.003	12,699.293	0.872	0.000	195 (85.2), 34 (14.8)
3	12,683.255	12,748.495	12,688.278	0.902	0.000	168 (73.4), 45 (19.6), 16 (7.0)
4	12,672.577	12,748.119	12,678.393	0.867	0.013	162 (70.7), 45 (19.7), 13 (5.7), 9 (3.9)
5	12,670.088	12,755.931	12,676.698	0.830	0.188	155 (67.7), 43 (18.8), 18 (7.9), 10 (4.4), 3 (1.3)

AIC = Akaike information criterion; BIC = Bayesian information criterion; SSABIC = sample-size adjusted BIC; BLRT Parametric Bootstrapped likelihood ratio test for K-1 (H0) vs. K classes.

**Table 4 ijerph-17-06973-t004:** Estimates of the Initial Value and the Rate of Change by Suicide Rate Change Pattern.

Model	Number of Cases (%)	Parameter Estimate		Change Types
		Initial Value	Rate of Change	
Model 1	168 (73.4)	33.964 ***	−1.072 ***	Average
Model 2	45 (19.6)	55.664 ***	−4.088 ***	Precipitous Drop
Model 3	16 (7.0)	40.152 ***	0.302	High Level

*** *p* < 0.001.

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
