# Peer review of "Regional Suicide Rate Change Patterns in Korea"

_ijerph, 2020, doi:10.3390/ijerph17196973_

Round 1

Reviewer 1 Report

This manuscript addresses an important topic, namely the occurrence of suicides and suicides patterns. My overall impression is that the manuscript is well-written, and the choice of method and variables in the conducted analysis is well-argued, too. In addition, the presentation of the results is clear and contributes to the field of suicide studies. I only have to minor comments/questions which the author(s) can addresses if they find them suitable or helpful for the manuscript.

Firstly, the result shows that that suicide tend to decrease in regions where the proportion of elderly residents are higher, and the number of social welfare facilities are higher (section 3.4), can this be an effect of that these regions has more facilities directed to elderly? Hence, the independent variable for mental health and welfare center include a variety of different facilities (section 2.2.2) and it can be hard to interpret if your result is on a general level (e.g. proportion of elderly residents and general welfare facilities), or that the result is dependent on specific facilities (if you want, this can be addressed in the discussion or in the section on limitations).

Secondly, in the discussion section (line 240 and forward in the manuscript) the authors discuss implementation of prevention, and the necessity to address local characteristics in preventive work. In this section it would be interesting if you elaborated on the importance of place and method of suicides in the preventive work, mainly since suicides tends to have a geographic pattern of “hotspots” and a pattern of “lethal methods” (see f.e. Stacks later work on suicide as a dramatic performance).

Reviewer 2 Report

This study examined suicide rate change patterns in South Korea and the predictive factors of suicide rate, which is interesting and helpful to inform prevention policies for Korean government. Overall, the paper is well written and easy to read, although it has a number of language errors. However, I think the authors still need to improve the quality of the paper as I have some concerns as follows:

  1. What is the theoretical framework for the study? Why did the authors study the relations of the proportion of elderly residents, number of social welfare facilities and etc. to suicide rate in Korea? The authors should explain more clearly about the rationale and theorectical framework for this research.
  2. The paper has numerous writing and linguistic errors, which needs further improvement. For example, Line 48 please use the full name of IMF; Line 50 "This study explores..." please use past tense here. It is best for the authors to check the entire paper and make corrections where needed.
  3. Section 2.1, it would be good if the authors could explain how representive the 229 cities, counties and districts chosen for the study.
  4. Line 88, what about the autonomous district? How was it coded in data analysis?
  5. Line 101,102, why didn't the authors use the number of mental health and welfare centers for data analysis? There should be some information missing if it is coded using 0 an 1.
  6. Section 2.3, the introduction to the data analyisis is too simple to understand. I think it is helpful to make it more specific to the audience.
  7. Unfortunately I am not familiar with GMM. However, I am currious why didn't the authors use regression analysis for the whole data set to determine the prediction of the independent variables.
  8. Lines 218-220, it is a bit arbitrary to conclude that "This indicates that the number of social welfare facilities 218 may be effective for supporting the needs of the elderly population, but not for the rest of the 219 population for decreasing the suicide rate." Maybe it is because the high levels of suicide rate in these areas, the local government built more social welfare facilities. I think the authors should explore and discuss why the number of social welfare facilities was associated with suicide rate differently across different regions, rather than making straightforward conclusion.

Reviewer 3 Report

I have read this paper with great interest, and applaud the project and effort as conducted. However, I would like to provide some additional suggestions to consider.  

Associations are not equal to causality. I therefor highly recommend to be more careful on the interpretation of the findings.

Abstract: if there is still word count left, I suggest to further extent the methods section of the abstract. In the results section, you use the word ‘distinct’ pattern, but this does not describe the trend of the pattern compared to the ‘average’, so I suggest to adapt wording.

Introduction

The introduction provides some overall data on suicide rate in South Korea, but some additional epidemiological data (like gender, age, how large is the overall population) are valuable to the external reader to better understand the subsequent sections. Along the same question, is the drop in suicide in all age categories ?

Methods

I applaud the analytical approach taken.

How confident are the authors on the data of the incidence on suicide ? (cfr abstract and methods), or why are the authors ‘sure’ that the data are reliable (no underreporting ?)

Is it correct to state that aspects like financial independence rate or the number of social welfare facilities have been assessed at the macro-level, but not to what extent these ‘interventions’ were active or not in the suicide cases (so based on average findings, not individual findings). Along this analysis, do you also have data on the ‘family’ structure or the ‘peers care’, or again, is this an analysis at the macro level ?  

Round 2

Reviewer 2 Report

The quality of the paper has been improved after this revision and I think it is ready for publication.